# OpenReview forum: "GSPR: Aligning LLM Safeguards as Generalizable Safety Policy Reasoners"
_ICLR.cc/2026/Conference — ICLR 2026 Conference Withdrawn Submission_

### Official Review · Reviewer_Nzi6 · 2025-10-24

**Soundness:** 3
**Presentation:** 3
**Contribution:** 2
**Rating:** 4
**Confidence:** 4

**Summary:**

This paper introduces GSPR (Generalizable Safety Policy Reasoner), a reinforcement-learning–aligned framework for fine-grained and generalizable LLM safety reasoning. The paper argues and proposes dynamically incorporating different safety policies from different providers to reason over various safety domains. GRPO is used in combination with SFT to improve reasoning quality and enforce structured otuput formatting.
Experimental results on eight benchmarks show that GSPR achieves state-of-the-art performance in both safety classification and category prediction, surpassing previous RL-aligned guardrails (e.g., RSafe) by up to 45% in category accuracy.

**Strengths:**

- Clear, reproducible training pipeline; ablations show each piece helps.
- Consistent S-Acc gains and meaningful C-Acc improvements in several settings.
- Practical attention to token cost and generation hygiene (repetition/language mixing).
- Useful qualitative cases; explanations improve transparency relative to pure classifiers.

**Weaknesses:**

* Novelty vs. Prior Multi-Policy Guardrails - There is a body of work training guardrails over multiple benchmarks/taxonomies and/or policy-conditioned prompts, and RL-aligned safety models. The paper needs a crisper positioning: what exactly is new beyond combining these ingredients? Ask: Add a thorough related-work section covering multi-policy/taxonomy conditioning and RL-aligned safety reasoners; run head-to-head with the strongest of those under identical taxonomies and format constraints.
* Stronger Baselines / Controls - Train baselines with the same flexible prompt (policy list + \safety/\category) but without GRPO (SFT only) and with PPO (no GRPO), matched for steps/compute, to isolate GRPO’s contribution.
* Include frontier API judge baselines with structured prompting for categories (where allowed), not just default settings.
* Robustness to Reward Hacking & Formatting - Add semantic judges (weak or learned reward models) to verify that correct categories are semantically justified, not just formatted.

**Questions:**

* There has been prior work on learning from various safety policies before, why not include in related work ? e.g GuardFormer : Guardrail Instruction
Pretraining for Efficient SafeGuarding - https://openreview.net/pdf?id=vr31i9pzQk
* How sensitive are results to α₁/α₂? Provide a sweep and show S-Acc/C-Acc/length trade-offs.
* What happens if you remove the format penalty but keep safety/category rewards?
* How does GSPR handle entirely novel policy text that overlaps semantically but not lexically with training categories?

---

> ### Author Response · Authors · 2025-11-15
> **Rebuttal by Authors (1/3)**
>
> We thank the reviewer for the valuable suggestions. Our responses are provided below.
>
> > Weakness 1. Novelty vs. Prior Multi-Policy Guardrails - There is a body of work training guardrails over multiple benchmarks/taxonomies and/or policy-conditioned prompts, and RL-aligned safety models. The paper needs a crisper positioning: what exactly is new beyond combining these ingredients? Ask: Add a thorough related-work section covering multi-policy/taxonomy conditioning and RL-aligned safety reasoners; run head-to-head with the strongest of those under identical taxonomies and format constraints.
>
> Thanks for pointing out the missing reference and we will properly discuss GuardFormer in the updated version. We have double-checked the related works and cannot find any work combining multiple taxonomies with fine-grained category-level evaluations (including GuardFormer). Most related works rely on their own safety taxonomies and only test binary safety evaluation on other benchmarks. GuardFormer considers the multi-taxonomy training but does not evaluate fine-grained category prediction performance and cannot provide any explanations. Additionally, GuardFormer is not open-sourced, and we cannot run head-to-head comparisons under identical taxonomies and format constraints.
>
> For evaluation, we use the exact and the same safety taxonomy of the evaluated benchmark and use this taxonomy to compare the performance with other baselines. We acknowledge that on certain benchmarks, such as Wildguard, GSPR may not be the best performing guardrail. Still, GSPR achieves the best averaged safe/unsafe prediction performance with unparalleled category prediction performance.
>
>
> > Weakness 2. Stronger Baselines / Controls - Train baselines with the same flexible prompt (policy list + \safety/\category) but without GRPO (SFT only) and with PPO (no GRPO), matched for steps/compute, to isolate GRPO’s contribution.
>
>
> Thanks for the suggestion. We have trained the following two baselines with the same flexible prompt for SFT-only and PPO. The out-of-domain results are as follows:
>
> |        | **Prompt Safety (↑)** |       |       |             |       |       | **Response Safety (↑)** |       |       |                   |       |       |             |       |
> | ------ | --------------------- | ----- | ----- | ----------- | ----- | ----- | ----------------------- | ----- | ----- | ----------------- | ----- | ----- | ----------- | ----- |
> |        | **Openai oderation**  |       |       | **HEx-PHI** |       |       | **T2T**                 |       |       | **Do-Not-Answer** |       |       | **Overall** |       |
> |        | S-Acc                 | S-F1  | C-Acc | S-Acc       | S-F1  | C-Acc | S-Acc                   | S-F1  | C-Acc | S-Acc             | S-F1  | C-Acc | S-Acc       | C-Acc |
> |**GSPR-Qwen2.5-SFT**              | 79.31                 | 70.07 | 76.42 | 96.89     | 98.42 | 33.10 | 98.98                  | 99.15 | 49.72 | 91.90           | 37.70 | 89.45 | 91.77       | 62.17 |
> |**GSPR-Qwen2.5-PPO**              | 86.15                 | 78.13 | 77.97 | 95.81     | 97.86 | 52.75 | 99.54                  | 99.61 | 85.75 | 92.48           | 36.69 | 89.24 | 93.50       | 76.43 |
> |**GSPR-Qwen2.5-GRPO**             | 83.42                 | 74.83 | 80.80 | 97.24     | 98.60 | 58.28 | 99.44                  | 99.53 | 90.05 | 91.26           | 38.81 | 89.67 | 92.84       | 79.70 |
>
> The results show that SFT-only performs poorly on both safe/unsafe and category prediction tasks, demonstrating the necessity of RL optimization to incentivize the guardrail's safety reasoning capabilities. On the other hand, PPO is comparable with GRPO improves the S-ACC at the cost of C-ACC. Therefore, the contribution of GSPR with GRPO is to balance both S-ACC and C-ACC for better trade-off, not dominating the overall performance.

---

> ### Author Response · Authors · 2025-11-15
> **Rebuttal by Authors (2/3)**
>
> > Weakness 3. Include frontier API judge baselines with structured prompting for categories (where allowed), not just default settings.
>
> Thanks for pointing out the missing baselines. We have tested gpt-4o-2024-08-06 and gpt-5-2025-08-07 with structured prompting for categories under the same evaluation setting. The results are as follows:
>
> |                  | **Prompt Safety (↑)** |       |       |           |       |       | **Response Safety (↑)** |       |       |                 |       |       |             |       |
> | ---------------- | --------------------- | ----- | ----- | --------- | ----- | ----- | ----------------------- | ----- | ----- | --------------- | ----- | ----- | ----------- | ----- |
> |                  | **Wildguard**         |       |       | **Aegis** |       |       | **SafeRLHF**            |       |       | **BeaverTails** |       |       | **Overall** |       |
> |                  | S-Acc                 | S-F1  | C-Acc | S-Acc     | S-F1  | C-Acc | S-Acc                   | S-F1  | C-Acc | S-Acc           | S-F1  | C-Acc | S-Acc       | C-Acc |
> | o3-mini          | 92.54                 | 90.96 | 80.34 | 58.53     | 70.71 | 37.88 | 86.87                   | 90.33 | 78.37 | 61.31           | 44.97 | 54.05 | 74.81       | 62.66 |
> | Gemini-2.5-Flash | 85.61                 | 83.95 | 72.54 | 63.27     | 75.48 | 47.28 | 85.05                   | 89.24 | 74.20 | 58.17           | 37.21 | 53.19 | 73.02       | 61.80 |
> | gpt-5-2025-08-07             | 99.76                 | 99.76 | 77.58 | 56.69     | 65.79 | 41.64 | 78.79                   | 82.50 | 64.90 | 54.56           | 0.83  | 54.56 | 72.45       | 59.67 |
> | gpt-4o-2024-08-06            | 85.93                 | 87.66 | 39.14 | 50.91     | 62.43 | 41.64 | 6.89                   | 78.88 | 82.45 | 53.25           | 0.20  | 13.94 | 68.71       | 22.44 |
> |**GSPR-Qwen2.5**              | 87.01                 | 83.70 | 72.57 | 82.99     | 78.10 | 79.04 | 90.32                  | 90.28 | 86.14 | 82.39           | 76.18 | 75.51 | 85.68       | 78.32 |
>
> These results suggest that even frontier API still struggles with the guardrail task, while our GSPR achieves the best performance.
>
>
> > Weakness 4. Robustness to Reward Hacking & Formatting - Add semantic judges (weak or learned reward models) to verify that correct categories are semantically justified, not just formatted.
>
>
> In results of Weakness 2, we experimented on PPO setup with a critic model to semantically judge the rollout trajectories of our actor model. The overall performance suggests that semantic judges perform similarly to formatting-based judges, which indicates that there is no significant reward hacking issue.
>
>
> > Q 1. There has been prior work on learning from various safety policies before, why not include in related work ? e.g GuardFormer : Guardrail Instruction Pretraining for Efficient SafeGuarding
>
> Thanks for pointing out the missing reference. We will properly discuss GuardFormer in the related works and experimental sections.
> As mentioned in Weakness 1, though GuardFormer also trained on various taxonomies, it still only considers evaluation on the binary safe/unsafe prediction. Our GSPR adopts fine-grained category prediction with explanations grounded on the applicable policies. Moreover, we adopt GRPO with tailored reward design and cold-start to further boost the safety reasoning performance.

---

> > ### Author Response · Authors · 2025-11-15
> > **Rebuttal by Authors (3/3)**
> >
> > > Q 2.How sensitive are results to α₁/α₂? Provide a sweep and show S-Acc/C-Acc/length trade-offs.
> >
> > Thanks for the suggestion. We have tested GSPR with different $\alpha_1$/$\alpha_2$ values and the results are as follows:
> >
> > |        | **Prompt Safety (↑)** |       |       |           |       |       | **Response Safety (↑)** |       |       |                 |       |       |             |       |
> > | ------ | --------------------- | ----- | ----- | --------- | ----- | ----- | ----------------------- | ----- | ----- | --------------- | ----- | ----- | ----------- | ----- |
> > |        | **Wildguard**         |       |       | **Aegis** |       |       | **SafeRLHF**            |       |       | **BeaverTails** |       |       | **Overall** |       |
> > |        | S-Acc                 | S-F1  | C-Acc | S-Acc     | S-F1  | C-Acc | S-Acc                   | S-F1  | C-Acc | S-Acc           | S-F1  | C-Acc | S-Acc       | C-Acc |
> > | $\alpha_0 = 0$, $\alpha_1 = 1$   | 87.65                 | 84.59 | 72.17 | 84.62     | 79.44 | 81.06 | 90.50                   | 90.09 | 85.69 | 82.60           | 74.98 | 76.03 | 86.34       | 78.74 |
> > | $\alpha_0 = 0.2$, $\alpha_1 = 0.8$ | 89.95                 | 88.17 | 72.87 | 83.56     | 79.29 | 79.18 | 89.74                   | 90.07 | 83.82 | 81.88           | 76.36 | 74.94 | 86.28       | 77.71 |
> > | $\alpha_0 = 0.4$, $\alpha_1 = 0.6$ | 90.26                 | 88.12 | 72.93 | 82.80     | 78.81 | 77.02 | 90.57                   | 90.42 | 85.20 | 83.06           | 78.16 | 75.70 | 86.67       | 77.71 |
> > | $\alpha_0 = 0.6$, $\alpha_1 = 0.4$ | 90.11                 | 88.23 | 73.87 | 84.13     | 79.49 | 79.53 | 91.25                   | 91.18 | 85.83 | 82.71           | 77.01 | 75.51 | 87.05       | 78.69 |
> > | $\alpha_0 = 0.8$, $\alpha_1 = 0.2$ | 89.64                 | 87.75 | 72.69 | 83.93     | 79.20 | 80.01 | 91.24                   | 91.08 | 86.42 | 82.17           | 75.82 | 75.46 | 86.74       | 78.65 |
> > | $\alpha_0 = 1$, $\alpha_1 = 0$   | 89.77                 | 87.89 | 63.98 | 84.09     | 79.57 | 75.07 | 91.30                   | 91.14 | 81.67 | 82.81           | 76.81 | 68.89 | 86.99       | 72.40 |
> >
> >
> > The results suggest that $\alpha_2$ (which corresponds to the factor of category reward) plays a more important role in boosting C-ACC, while $\alpha_1 = 1, \alpha_2 = 0 $ reduces C-ACC significantly. Therefore, our reward design is more curical than RSafe.
> >
> >
> > > Q 3. What happens if you remove the format penalty but keep safety/category rewards?
> >
> > It does not make sense to fully remove the format penalty. The format penalty stabilizes learning by enforcing the desired output formats. It avoids unparseable or malformed responses during RL rollouts. Without it, it is impossible to train the LLM via rule-based rewards for RL. Only under the correct format, the accuracy rewards will be assigned to the rollouts.
> >
> >
> > In Table 3, we can see that the repetition check and code switching check are not applied on RSafe and GSPR (safety only). Both settings incur a significant increase in language mixing and repetition ratios. This partially explains the importance of format checking.
> >
> >
> > > Q 4. How does GSPR handle entirely novel policy text that overlaps semantically but not lexically with training categories?
> >
> > Regardless of the policy similarity, the required policies will be put into the prompt templates. If within a taxonomy, one unsafe sample may correspond to more than 1 category due to policies’ semantic overlap. GSPR will output only 1 category that best explains the sample’s unsafe behavior. One violated category is sufficient to mark this sample as unsafe.

---

> ### Author Response · Authors · 2025-11-26
>
> Dear Reviewer Nzi6,
>
> We want to thank you again for your comments and suggestions. They have been helpful in improving the quality of our paper.
>
> We would like to know if our responses have addressed your concerns. If you have any further questions, please let us know.
>
> Thanks a lot!

---

### Official Review · Reviewer_EdSy · 2025-10-28

**Soundness:** 2
**Presentation:** 3
**Contribution:** 1
**Rating:** 4
**Confidence:** 3

**Summary:**

This work addresses the challenge that existing reasoning-based guard models struggle with fine-grained safety category prediction. To improve category-wise safety moderation, the authors propose GSPR, which introduces variable and flexible safety taxonomies into the prompt rather than using a fixed set of policies. The model is trained using GRPO similarly to RSafe. Experiments demonstrate meaningful improvements in category classification accuracy compared to prior methods.

**Strengths:**

- The paper clearly introduces the importance of accurate category classification beyond simple safe/unsafe decisions in real-world LLM safety applications. This focus on explainability and policy grounding is well motivated.
- The proposed GSPR shows notable empirical gains in category prediction performance over existing guard models and training strategies. This suggests that prompting with flexible safety categories is an effective design choice for improving guardrail reasoning.

**Weaknesses:**

##

- While the proposed approach is practical and performs well, the methodological novelty may feel somewhat incremental. Reinforcement learning for guardrail alignment has been previously explored (e.g., RSafe [1]), and introducing dynamic categories in prompts could be seen as a natural step when optimizing for fine-grained categorization. Clarifying the conceptual novelty and broader implications could further strengthen the contribution.
- In Table 2, the improvement in safety accuracy (S-Acc) with Cold-Start is promising. However, since the Cold-Start stage uses carefully curated data from Gemini-2.5-Flash and GSPR w/o Cold-Start performs similarly to RSafe, one could reasonably expect RSafe to benefit similarly from a comparable warm-start strategy. Although the Cold-Start itself is useful, its originality may be limited given its prevalence in RL-based alignment pipelines [2]. Providing more justification for its role or demonstrating alternatives would help highlight the contribution.

[1] Zheng et al., RSafe: Incentivizing proactive reasoning to build robust and adaptive LLM safeguards

[2] DeepSeek-AI, DeepSeek-R1: Incentivizing Reasoning Capability in LLMs via Reinforcement Learning

**Questions:**

- For baselines that rely on fixed safety policies, how are these policies incorporated into prompts? Since different benchmarks define different taxonomies, does each dataset use a separate prompt design, or is a unified prompt format used (e.g., all policies)? It would be helpful to clarify whether an ablation exists.
- At inference time, how are the safety categories selected for the prompt? Are benchmark-specific categories provided, or are all categories included to ensure full coverage?
- In practical deployment, the model deployers may not know in advance which taxonomy applies. How would the system operate in real-world scenarios? Would category retrieval or selection be necessary to determine which policies to present to the model?

---

> ### Author Response · Authors · 2025-11-15
> **Rebuttal by Authors (1/2)**
>
> We thank the reviewer for the valuable suggestions. Our responses are provided below.
>
>
> > Weakness 1 While the proposed approach is practical and performs well, the methodological novelty may feel somewhat incremental. Reinforcement learning for guardrail alignment has been previously explored (e.g., RSafe [1]), and introducing dynamic categories in prompts could be seen as a natural step when optimizing for fine-grained categorization. Clarifying the conceptual novelty and broader implications could further strengthen the contribution.
>
> We agree that reinforcement learning and guardrail alignment have been explored previously (e.g., RSafe) and it is natural to introduce dynamic categories in prompts. Our GSPR takes this nature step and our contributions are listed below:
>
>
> - **Safety reasoning grounded on varied and well-annotated taxonomies.** GSPR introduces a distinct conceptual advance by explicitly reasoning around various expert-annotated, fine-grained taxonomies that go beyond dynamic categories.  Only considering dynamic categories would likely introduce inconsistency, ambiguity, or even conflicting safety judgments. Consequently, these dynamic policies may undermine the safety reasoning performance. GSPR enables interpretable, consistent, and domain-aware safety judgments across diverse taxonomies.
>
> - **Careful safety reward design with cold-start strategies.** As shown in Table 4, our GSPR greatly reduces the code-switching (language mixing) and repetition ratios on the Qwen base models, especially on Qwen 3. Therefore, our reward design greatly improves the quality of reasoning traces and the cold-start process stabilizes the model at the early stage of training.
>
>
> - **Better inference efficiency.**  Our GSPR significantly reduces the token cost to be the most efficient reasoning guardrail LLM.
>
>
> In summary, through extensive experiments, our GSPR provides a practical and generalizable recipe for training a robust guardrail LLM through RL. GSPR offers explainable and policy-consistent safety judgments, which are crucial for real-world safety-critical applications. Unlike prior works, GSPR generates its reasoning traces grounding on the exact policies, improving its confidence to draw the final safety decisions.
>
>
> > Weakness 2 In Table 2, the improvement in safety accuracy (S-Acc) with Cold-Start is promising. However, since the Cold-Start stage uses carefully curated data from Gemini-2.5-Flash and GSPR w/o Cold-Start performs similarly to RSafe, one could reasonably expect RSafe to benefit similarly from a comparable warm-start strategy. Although the Cold-Start itself is useful, its originality may be limited given its prevalence in RL-based alignment pipelines [2]. Providing more justification for its role or demonstrating alternatives would help highlight the contribution.
>
>
> Compared with Rsafe, a careful cold-start process is one of GSPR’s key contribution to train a guardrail LLM via RL. Without it, the model would be exposed to low-quality rollout trajectories, which can lead to unstable training and degraded safety reasoning performance. Our proposed cold-start specifically bootstraps fine-grained safety reasoning using existing foundation models, ensuring consistent, interpretable judgments and improving the effectiveness of RL optimization.
>
> In addition, even without cold-start, our GSPR still significantly outperforms RSafe on C-ACC by more than 50\% and achieves comparable S-ACC, demonstrating the effectiveness of our overall design.
>
> > Question 1 For baselines that rely on fixed safety policies, how are these policies incorporated into prompts? Since different benchmarks define different taxonomies, does each dataset use a separate prompt design, or is a unified prompt format used (e.g., all policies)? It would be helpful to clarify whether an ablation exists.
>
>
> For baselines that rely on fixed safety policies, such as Llama-Guard and ShieldGemma, if we evaluate C-Acc for the baselines, we follow the baselines’ formatting and change their policies inside their prompts. Then, we implement an LLM-specific parser to capture models’ predictions following their original output patterns.
> For different taxonomies, we only use their covered policies for training, not putting all policies into the prompt. Putting all policies into the prompt would likely introduce confusion and inconsistency.

---

> > ### Author Response · Authors · 2025-11-15
> > **Rebuttal by Authors (2/2)**
> >
> > > Question 2 At inference time, how are the safety categories selected for the prompt? Are benchmark-specific categories provided, or are all categories included to ensure full coverage?
> >
> >
> > Yes, benchmark-specific categories are provided. For running testing via inference, each testing data point is associated with its safety taxonomy, and we use this taxonomy to evaluate the performance.
> >
> > > Question 3 In practical deployment, the model deployers may not know in advance which taxonomy applies. How would the system operate in real-world scenarios? Would category retrieval or selection be necessary to determine which policies to present to the model?
> >
> > During inference, the developers may input their own defined safety taxonomies for the guardrail tasks, which fits into our out-of-domain senario. We also set the Aegis as the default safety taxonomy if no category is provided. We intentionally do not want to put all categories together, as they do not account for specific downstream use cases.

---

> ### Author Response · Authors · 2025-11-26
>
> Dear Reviewer EdSy,
>
> We want to thank you again for your comments and suggestions. They have been helpful in improving the quality of our paper.
>
> We would like to know if our responses have addressed your concerns. If you have any further questions, please let us know.
>
> Thanks a lot!

---

### Official Review · Reviewer_Np2S · 2025-10-31

**Soundness:** 2
**Presentation:** 2
**Contribution:** 2
**Rating:** 2
**Confidence:** 4

**Summary:**

This paper introduces GSPR, a training pipeline aimed at enabling reasoning models to detect unsafe input prompts and outputs. A central claimed contribution is that the algorithm facilitates adaptation to diverse safety taxonomies through a deliberate cold-start strategy and targeted reward engineering. The authors also conduct experiments to evaluate the GSPR's efficacy.

**Strengths:**

1. The motivation is reasonable, and the field is important. Safeguards for large language models (LLMs) must indeed support the generalization of flexible safety policies.
2. The authors address unseen safety policies and evaluate the conditions under which adaptation occurs.

**Weaknesses:**

1. **Limited Novelty**. From my perspective, this paper does not introduce substantial innovations. It primarily applies GRPO—a well-established algorithm—to safety evaluation tasks and reasoning models. The highlighted motivation and contributions strike me as somewhat trivial. A capable LLM can already adapt to varying safety policies embedded in prompts, leveraging its strong instruction-following capabilities.
2. The paper's use of distillation from Gemini Flash for cold-start initialization only reinforces this view. I suspect that a sufficiently powerful model could handle this task effectively without such interventions. Moreover, the task design itself raises concerns: In an era of highly advanced models, why constrain the candidate policies to a mere 167 examples? Why not generate rubrics dynamically or retrieve them from a much larger pool of thousands?
3. Building on the above, the scale of the 167 policies feels underwhelming and less meaningful than it could be. In my view, the true crux of success here lies in the LLM's instruction-following prowess. I would wager that these policies are fairly commonplace and already ingrained in modern LLMs such as Qwen, DeepSeek, and LLaMA—even in your so-called "out-of-domain" safety evaluations. Have the authors considered testing with truly uncommon or even counterfactual policies? Would the model degrade under such conditions, or maintain robustness?
4. The authors overlook several critical baselines, especially those from GenRM literature. Notably, the safety tasks targeted here represent only a narrow subset of broader reward modeling (RM) challenges. Many RM datasets (e.g., Skywork-Reward-Preference and your used wildguard) already encompass safety classification. Furthermore, some reward models can autonomously generate flexible policies (or rubrics) before applying judgments. In contrast, this work's format remains constrained: most prompts are static, with only minor flexibility introduced. This limits its scope compared to more dynamic approaches. Numerous prior studies have tackled analogous challenges, yet this paper fails to discuss them meaningfully. Relevant examples include:

[1] [ShieldLM: Empowering LLMs as Aligned, Customizable and Explainable Safety Detectors](https://arxiv.org/pdf/2402.16444)

[2] [Inference-Time Scaling for Generalist Reward Modeling](https://arxiv.org/pdf/2504.02495)

[3] [J1: Incentivizing Thinking in LLM-as-a-Judge via Reinforcement Learning](https://arxiv.org/pdf/2505.10320)

[4] [RM-R1: Reward Modeling as Reasoning](https://arxiv.org/pdf/2505.02387)

[5] [Qwen3Guard Technical Report](https://arxiv.org/pdf/2510.14276v1) (very recently published; could be excused if overlooked))

In summary, the use of dynamic rubrics or principles is now commonplace in this community. Against this backdrop, I find the paper's contributions insufficiently distinctive.

**Questions:**

See above.

---

> ### Author Response · Authors · 2025-11-15
> **Rebuttal by Authors (1/3)**
>
> We thank the reviewer for the valuable suggestions. We believe there are a lot of misunderstandings about the current review. Our responses are provided below.
>
> > Weakness 1 Limited Novelty. The highlighted motivation and contributions strike me as somewhat trivial. A capable LLM can already adapt to varying safety policies embedded in prompts, leveraging its strong instruction-following capabilities.
>
> We believe there are a few misunderstandings about our novelty and contributions. Compared with existing strong instruction-following LLMs, the contributions of GSPR are listed below:
> - Better guardrail performance with fewer post-training data and a smaller model size. As shown in Tables 2 and 3, our GSPR consistently achieves the best averaged performance for both in-domain and out-of-domain settings. Compared with closed-source proprietary LLMs, our GSPR can be trained on small-scale LLMs and approximately 10,000 - 20,000 training samples.
> - Superior performance on fine-grained violated safety policy prediction. Regarding the fine-grained policy prediction task, our GSPR achieves the best C-ACC with more than 10\% improvements.
> - Careful safety reward design with proper cold-start strategy. As shown in Table 4, our GSPR greatly reduces the code-switching (language mixing) and repetition ratios on the Qwen base models, especially on Qwen 3. Therefore, our reward design greatly improves the quality of reasoning traces.
> - The best inference efficiency among reasoning guardrails. Our GSPR significantly reduces the token cost to be the most efficient reasoning guardrail LLM.
>  Therefore, our GSPR is definitely non-trivial and achieves the best safety reasoning performance with the best efficiency.
>
>
>
>
> > Weakness 2 The paper's use of distillation from Gemini Flash for cold-start initialization only reinforces this view. I suspect that a sufficiently powerful model could handle this task effectively without such interventions. Moreover, the task design itself raises concerns: In an era of highly advanced models, why constrain the candidate policies to a mere 167 examples? Why not generate rubrics dynamically or retrieve them from a much larger pool of thousands?
>
> We appreciate the reviewer’s comments and would like to clarify several points.
>
> - First, our use of distillation from Gemini Flash is not a dependency but a cold-start initialization to stabilize the model at the early stage of training.  This approach improves training performance without limiting GSPR’s capability to generalize.
>
>
> - Second, the “167 policies” used in the paper refer to unique safety policies aggregated from nearly all existing open-source safety benchmarks, not the number of training samples. As shown in Tables 5 and 6, our collected corpus actually exceeds **500,000** samples, and we sample approximately 20,000 examples from each dataset to train GSPR. We would like to clarify that our GSPR can be trained to any new safety taxonomies with specified new policies.
>
> - Third, compared with generated rubrics, **our training policies are carefully curated from expert-annotated safety taxonomies.**  Each taxonomy defines a coherent and non-contradictory set of policies covering a particular domain to serve its complex safety purpose. Dynamically generating rubrics or retrieving them from a massive pool would likely introduce inconsistency, ambiguity, or even conflicting safety judgments. Consequently, these generated rubrics may undermine the safety reasoning performance.
>
> In summary, though dynamic rubric generation looks suitable for open-ended tasks such as instruction following evaluation, safety evaluation requires taxonomies grounded in expert knowledge and consistent policy definitions, as exemplified by frameworks such as OpenAI’s policies and NVIDIA’s AEGIS. GSPR is designed to flexibly adapt to any such taxonomy while preserving the necessary rigor and interpretability for safety-critical assessment.
>
> Here we also test existing strong foundation models on the fine-grained safety policy prediction task and the results are as follows:

---

> ### Author Response · Authors · 2025-11-15
> **Rebuttal by Authors (2/3)**
>
> |                  | **Prompt Safety (↑)** |       |       |           |       |       | **Response Safety (↑)** |       |       |                 |       |       |             |       |
> | ---------------- | --------------------- | ----- | ----- | --------- | ----- | ----- | ----------------------- | ----- | ----- | --------------- | ----- | ----- | ----------- | ----- |
> |                  | **Wildguard**         |       |       | **Aegis** |       |       | **SafeRLHF**            |       |       | **BeaverTails** |       |       | **Overall** |       |
> |                  | S-Acc                 | S-F1  | C-Acc | S-Acc     | S-F1  | C-Acc | S-Acc                   | S-F1  | C-Acc | S-Acc           | S-F1  | C-Acc | S-Acc       | C-Acc |
> | o3-mini          | 92.54                 | 90.96 | 80.34 | 58.53     | 70.71 | 37.88 | 86.87                   | 90.33 | 78.37 | 61.31           | 44.97 | 54.05 | 74.81       | 62.66 |
> | Gemini-2.5-Flash | 85.61                 | 83.95 | 72.54 | 63.27     | 75.48 | 47.28 | 85.05                   | 89.24 | 74.20 | 58.17           | 37.21 | 53.19 | 73.02       | 61.80 |
> | gpt-5-2025-08-07             | 99.76                 | 99.76 | 77.58 | 56.69     | 65.79 | 41.64 | 78.79                   | 82.50 | 64.90 | 54.56           | 0.83  | 54.56 | 72.45       | 59.67 |
> | gpt-4o-2024-08-06            | 85.93                 | 87.66 | 39.14 | 50.91     | 62.43 | 41.64 | 6.89                   | 78.88 | 82.45 | 53.25           | 0.20  | 13.94 | 68.71       | 22.44 |
> |**GSPR-Qwen2.5**              | 87.01                 | 83.70 | 72.57 | 82.99     | 78.10 | 79.04 | 90.32                  | 90.28 | 86.14 | 82.39           | 76.18 | 75.51 | 85.68       | 78.32 |
>
> These results further demonstrate that existing strong foundation models still struggle with fine-grained safety policy prediction, while our GSPR achieves the best performance.
>
>
> > Weakness 3 Building on the above, the scale of the 167 policies feels underwhelming and less meaningful than it could be. In my view, the true crux of success here lies in the LLM's instruction-following prowess. I would wager that these policies are fairly commonplace and already ingrained in modern LLMs such as Qwen, DeepSeek, and LLaMA—even in your so-called "out-of-domain" safety evaluations. Have the authors considered testing with truly uncommon or even counterfactual policies? Would the model degrade under such conditions, or maintain robustness?
>
>
> We respectfully **disagree that the number of policies is the key determinant of GSPR’s contribution**. Though Instruction-following indeed helps model for the guardrail task,  it alone cannot account for robust guardrail behavior, as safety awareness does not come from instruction following. As shown in previous Table in Weakness 2, the LLMs’ safety reasoning also plays a key role in the guardrail task. Our GSPR leverages several expert-annotated taxonomies to enhance LLMs’ safety reasoning abilities with fine-grained taxonomies.
>
>
> Moreover, the reviewer’s concern about "out-of-domain" setup is addressed by the temporal separation. Specifically, Qwen2.5’s knowledge cutoff is October 2023, whereas the majority of our training and evaluation data were curated after 2024. Therefore, these taxonomies and examples are not memorized by the underlying base model.
>
> Regarding truly uncommon or counterfactual policies, **we would like to emphasize that expert-developed taxonomies are intentionally designed to reflect realistic and semantically valid safety standards.** Counterfactual or arbitrary policies often lack coherent safety semantics and thus fall outside the intended scope of rigorous safety evaluation.

---

> > ### Author Response · Authors · 2025-11-15
> > **Rebuttal by Authors (3/3)**
> >
> > > Weakness 4 The authors overlook several critical baselines, especially those from GenRM literature. Notably, the safety tasks targeted here represent only a narrow subset of broader reward modeling (RM) challenges. Many RM datasets already encompass safety classification. Furthermore, some reward models can autonomously generate flexible policies (or rubrics) before applying judgments. In contrast, this work's format remains constrained: most prompts are static, with only minor flexibility introduced. This limits its scope compared to more dynamic approaches. Numerous prior studies have tackled analogous challenges, yet this paper fails to discuss them meaningfully.
> >
> >
> >
> > We thank the reviewer for pointing out the additional related works. We will properly discuss these works for the updated version.
> >
> >
> > While we recognize that some reward modeling frameworks address safety to a certain extent, **our GSPR differs in both goal and granularity.** GSPR is not designed as a general preference model but as a fine-grained safety reasoning guardrail that reasons on model behavior with given taxonomies rather than scalar preferences. Consequently, most GenRM cannot be used in this case.
> >
> > Moreover, unlike prior RM approaches that focus primarily on preference alignment or dynamic rubric generation, GSPR emphasizes interpretable and taxonomy-consistent safety judgments to **exactly pin-point the associated category.** In practice, the safety rules are already given, such as safety policies and regulations. Our design ensures reproducible and transparent reasoning across safety domains, which is crucial for guardrail applications and future safety annotations.
> >
> > Additionally, for the task of safety moderation, we only care about the moderation performance. **Static templates are widely used to boost the task performance for nearly all the guardrails, including gpt-oss-safeguard, qwen3-guard, and even your mentioned  ShieldLM.** It makes no sense to care about moderation format flexibility.
> >
> > Lastly, in terms of overall performance, our evaluation shows that GSPR achieves strong safety performance and reasoning robustness, which dominates general RM models such as WildGuard.

---

> ### Author Response · Authors · 2025-11-26
>
> Dear Reviewer Np2S,
>
> We want to thank you again for your comments and suggestions. They have been helpful in improving the quality of our paper.
>
> We would like to know if our responses have addressed your concerns. If you have any further questions, please let us know.
>
> Thanks a lot!

---

### Official Review · Reviewer_riCS · 2025-11-01

**Soundness:** 3
**Presentation:** 3
**Contribution:** 2
**Rating:** 4
**Confidence:** 4

**Summary:**

The paper proposes GSPR, a Generalizable Safety Policy Reasoner to identify unsafe input prompts and LLMs’ outputs with violated safety taxonomies through Group Relative Policy Optimization (GRPO). The results suggest that it can achieve remarkable detection performance across various safety benchmarks.

**Strengths:**

- Comprehensive policy coverage. It includes 19 fine-grained taxonomies with 167 safety policies.
- Strong performances. The method demonstrates strong performance over the baselines.

**Weaknesses:**

- Missing related works. Some related works also consider fine-grained safety classification beyond binary classification, such as ShieldLM and Qwen3Guard.
- Lack of in-depth analysis. We can only see that the method performs well across different benchmarks. But how safety policies are different across these benchmarks are unclear. For example, will the same sample be safe under some policy but be unsafe under another policy? It is necessary to conduct in-depth analysis regarding the concrete role of safety policies.
- Lack of ablation study on the training data. The paper mixes data from various sources for training, but the impact of these training data from different sources is unclear.

**Questions:**

None

---

> ### Author Response · Authors · 2025-11-15
> **Rebuttal by Authors (1/2)**
>
> > Weakness 1 Missing related works. Some related works also consider fine-grained safety classification beyond binary classification, such as ShieldLM and Qwen3Guard.
>
> Thanks for pointing out the missing related works. We will properly cite and discuss these two works in the updated paper. We agree with the reviewer that fine-grained safety standards are widely explored and we have cited many of these related works. However, we would like to clarify a few misunderstandings.
>
>
> - First, Qwen3Guard is arxived behind the ICLR submission deadline, and we cannot discuss papers that show up later than the submission deadline. That is, Qwen3 Guard is a concurrent work.
>
>
> - Second, some related works do consider fine-grained safety classification beyond binary classification (e.g., by including “controversial” cases and rule-based distinctions), but their evaluations remain limited in scope. For example,  ShieldLM is customizable to detection rules, but its core evaluation and dataset seem centered on one annotation scheme and focus mostly on determining safe/unsafe/controversial rather than addressing the problem of identifying which rules or categories are applicable in a given context. Instead, the key contribution of GSPR is to provide better support for multiple taxonomies and varied safety policies with fine-grained category prediction that identifies which exact category is violated.
>
> Consequently, our GSPR naturally generalizes better to new safety taxonomies by design.
>
>
> > Weakness 2 Lack of in-depth analysis. We can only see that the method performs well across different benchmarks. But how safety policies are different across these benchmarks are unclear. For example, will the same sample be safe under some policy but be unsafe under another policy? It is necessary to conduct in-depth analysis regarding the concrete role of safety policies.
>
> For different benchmarks, different taxonomies may apply to their samples. It’s possible that the same sample may be labeled differently according to the applied policies. For example, in OpenAI’s safety policies, users cannot use ChatGPT for “provision of tailored advice that requires a license, such as legal or medical advice, without appropriate involvement by a licensed professional.” Regarding medical advice on reducing blood pressure, it should be prohibited, given OpenAI’s taxonomy. However, it is regarded as safe for other taxonomies that do not have restrictions on medical advice, such as WildGuard.

---

> > ### Author Response · Authors · 2025-11-15
> > **Rebuttal by Authors (2/2)**
> >
> > > Weakness 3 Lack of ablation study on the training data. The paper mixes data from various sources for training, but the impact of these training data from different sources is unclear.
> >
> > We hope the reviewer could excuse the limited computational resources at the university. We leverage 22 taxonomies to train the GSPR, and it is hard to fully perform ablation studies on all the data.
> > Still, we manage to train GSPR under two different sets of training data and the results are as follows:
> >
> > |                                                         | **Prompt Safety (↑)** |       |       |           |       |       |                       |       |       |             |       |       | **Response Safety (↑)** |       |       |                 |       |       |         |       |       |                   |       |       |             |       |
> > | :-----------------------------------------------------: | --------------------- | ----- | ----- | --------- | ----- | ----- | --------------------- | ----- | ----- | ----------- | ----- | ----- | ----------------------- | ----- | ----- | --------------- | ----- | ----- | ------- | ----- | ----- | ----------------- | ----- | ----- | ----------- | ----- |
> > |                                                         | **Wildguard**         |       |       | **Aegis** |       |       | **Openai Moderation** |       |       | **HEx-PHI** |       |       | **SafeRLHF**            |       |       | **BeaverTails** |       |       | **T2T** |       |       | **Do-Not-Answer** |       |       | **Overall** |       |
> > |                                                         | S-Acc                 | S-F1  | C-Acc | S-Acc     | S-F1  | C-Acc | S-Acc                 | S-F1  | C-Acc | S-Acc       | S-F1  | C-Acc | S-Acc                   | S-F1  | C-Acc | S-Acc           | S-F1  | C-Acc | S-Acc   | S-F1  | C-Acc | S-Acc             | S-F1  | C-Acc | S-Acc       | C-Acc |
> > |           **GSPR (base on aegis+wildguard)**            | 86.82                 | 84.49 | 69.98 | 81.06     | 76.67 | 75.70 | 84.24                 | 75.62 | 81.82 | 95.52       | 97.71 | 44.48 | 85.97                   | 84.72 | 79.66 | 77.92           | 67.27 | 68.51 | 95.86   | 96.46 | 90.70 | 91.05             | 33.33 | 89.35 | 87.31       | 75.03 |
> > | **GSPR (base on aegis+wildguard+SafeRLHF+BeaverTails)** | 86.58                 | 83.59 | 69.51 | 82.43     | 77.82 | 77.72 | 84.96                 | 76.07 | 83.11 | 95.86       | 97.89 | 48.62 | 88.46                   | 87.52 | 83.78 | 80.24           | 70.81 | 74.23 | 97.93   | 98.24 | 91.11 | 92.44             | 38.26 | 91.16 | 88.61       | 77.41 |
> > |             **GSPR (base on all datasets)**             | 87.01                 | 83.70 | 72.57 | 82.99     | 78.10 | 79.04 | 83.42                 | 74.83 | 80.80 | 97.24       | 98.60 | 58.28 | 90.32                   | 90.28 | 86.14 |82.39           | 76.18 | 75.51 | 99.44   |99.53 | 90.05 | 91.26             | 38.81 | 89.67 | 89.60       | 78.80 |
> >
> >
> > From the results, we can see that adding more training data from varied sources generally improves the overall performance of GSPR across different benchmarks for both in-domain and out-of-domian setups. This demonstrates the effectiveness of leveraging diverse safety taxonomies for training a more robust and generalizable LLM guardrail.

---

> > > ### Author Response · Authors · 2025-11-26
> > >
> > > Dear Reviewer riCS,
> > >
> > > We want to thank you again for your comments and suggestions. They have been helpful in improving the quality of our paper.
> > >
> > > We would like to know if our responses have addressed your concerns. If you have any further questions, please let us know.
> > >
> > > Thanks a lot!

---

### Note · Authors · 2026-01-04

I have read and agree with the venue's withdrawal policy on behalf of myself and my co-authors.